

**Use of an Uncrewed Aerial System to Investigate Aerosol Direct and Indirect Radiative**
**Forcing Effects in the Marine Atmosphere**
Patricia K. Quinn[1], Timothy S. Bates[2], Derek J. Coffman[1], James E. Johnson[2], and Lucia M.
Upchurch[2]
[1]NOAA Pacific Marine Environmental Laboratory, Seattle, WA 98115, USA
[2]University of Washington Cooperative Institute for Climate, Ocean, and Ecosystem Studies,
Seattle, WA 98105, USA
*Correspondence to*: Patricia K. Quinn (patricia.k.quinn@noaa.gov)
**Abstract**
An uncrewed aerial system (UAS) has been developed for observations of aerosol and cloud properties relevant to
aerosol direct and indirect forcing in the marine atmosphere. The UAS is a hybrid quadrotor – fixed wing aircraft
designed for launch and recovery from a confined space such as a ship deck. Two payloads, Clear Sky and Cloudy
Sky, house instrumentation required to characterize aerosol radiative forcing effects. The observing platform (UAS
plus payloads) has been deployed from a ship and from a coastal site for observations in the marine atmosphere. We
describe here the details of the UAS, the payloads, and first observations from the *TowBoatUS Richard L. Becker*
(March 2022) and from the Tillamook UAS Test Range (August 2022).
**1. Introduction**
Atmospheric aerosol particles affect the Earth's radiation budget directly by scattering and absorbing incoming solar
radiation and indirectly by taking up water and forming cloud droplets. Chemical composition of the particles
determines, in part, whether they scatter incoming solar radiation back to space which leads to cooling at the Earth's
surface or absorb radiation and warm layers within the atmosphere (e.g., Li et al., 2022). The amount of heating
depends on the vertical distribution of the absorbing aerosol layer, whether it is located above or below clouds, and
the albedo of the surface (Takemura et al., 2002; Haywood and Ramaswamy, 1998). Whether particles act as cloud
condensation nuclei (CCN) and nucleate cloud droplets depends on their size and chemical composition (Lohmann
and Feichter, 2005). If the particles are large enough and contain sufficient soluble material, an increase in particle
number can lead to an increase in cloud droplet number concentration and cloud albedo thereby leading to a cooling
at the Earth's surface. The degree to which aerosol direct and indirect forcing are cooling the planet and offsetting
warming by greenhouse gases is highly uncertain. According to the International Panel on Climate Change (IPCC),
aerosols contribute the largest uncertainty in quantifying present-day climate change (IPCC, 2021).
Vertical profiles of aerosol and cloud properties are required to improve models and decrease uncertainties,
particularly over oceans due to the susceptibility of marine clouds to small changes in aerosol concentrations
(Rosenfeld et al., 2019). While satellite observations have the advantage of providing global coverage, *in situ*
observations have the highest level of accuracy available to constrain radiative forcing and reduce uncertainties in



forcing estimates (Li et al., 2022). Crewed aircraft have been used for the past several decades to characterize
horizontal and vertical distributions of aerosol and cloud properties relevant to radiative forcing (e.g., Russell et al.,
1999; Yoon and Kim, 2006; Zhang et al., 2017). These measurements come at a relatively high cost and require
extensive logistical coordination.

Uncrewed aerial systems, or UAS, have the advantage of lower costs and flexibility and frequency of flights
compared to crewed aircraft. In addition, they offer higher spatial resolution due to their relatively slow flight speed.
UAS have been used since the mid-2000s to measure aerosol and cloud properties relevant to radiative forcing
including particle number concentration and size distribution, light absorption, aerosol optical depth, and cloud drop
number and effective radius. These measurements have been made with vertical-take-off-and landing (VTOL) UAS,
either quadcopters (Brady et al., 2016) or hexacopters (e.g., Chilinski et al., 2016; Aurell et al., 2017), or fixed wing
UAS (Corrigan et al., 2008; Bates et al., 2013). The VTOL UAS that have been used have the advantage of not
needing a catapult or runway to be launched and recovered but typically have short endurance (< 30 min) and a
limited altitude ceiling (~1 km). The fixed wing aircraft that have been used require a launch and recovery apparatus
or a runway but have the advantage of longer duration (hours) and can reach high altitudes of 3 km or more. While
some VTOLs used can carry relatively heavy payloads (6 kg or more), they can only do so for ~ 15 min while some
of the fixed wing UAS can carry heavy payloads for hours.

We report here on measurements of aerosol and cloud properties using a hybrid quadrotor – fixed wing UAS, the
Fixed Wing VTOL Rotator or FVR-55, developed by L3Harris Latitude Engineering. The hybrid quadrotor – fixed
wing concept combines the advantages of fixed wing flight with the ability to take-off and land vertically thus
eliminating the need for a runway and allowing for launch and recovery from ships and other confined spaces. The
FVR-55 has an endurance of more than 4 hours, a height ceiling of at least 3 km, and can carry a 6 kg payload.
NOAA PMEL has developed two UAS payloads -- one for the measurement of aerosol properties relevant to direct
radiative forcing (Clear Sky) and one for the measurement of aerosol and cloud properties relevant to indirect
forcing (Cloudy Sky). The FVR-55 and instrumentation in the two payloads are described herein along with the
results of its first shipboard and coastal flights.

**2. Methods**

**2.1. FVR-55**

The FVR-55, a class II medium endurance UAS, was developed by Latitude Engineering (since acquired by
L3Harris) with support from NOAA Phase I and II SBIRs (Small Business Innovation Research awards). The focus
of the SBIR was a UAS able to carry a 5.5 kg payload, have a flight ceiling of up to 3 km, an endurance of 3 hrs or
more, and a pusher engine. The hybrid quadrotor – fixed wing design of the FVR-55 combines the high-power
density of electric motors and propellers with the high-energy density of a piston engine and liquid fuel. The



electric-quadrotor system is used during launch and recovery (high power, short endurance) and the gas engine is
used for fixed wing flight (low power, long endurance). The aircraft has an empty weight of 20 kg and a maximum
take-off weight of 29.5 kg. It cruises at 25 m s$^{-1}$. A Cloud Cap Piccolo autopilot flight controller is used for
autonomous flight. In the case of a lost link, the avionics guides the UAS to a predetermined return to base location
and, if communication is not re-established, to land at an established target. A mobile ground control station
(Windows tablet or Laptop with Datalink) provides ground command and control. A differential GPS (dGPS)
system is used for computing the aircraft's heading to circumvent the challenges created by the hull of a ship
distorting the Earth's magnetic field. The fuselage of the FVR-55 was designed for a maximum spacing of the two
dGPS antennas to increase the accuracy of the computed heading. VTOL motors and propellers provide enough
overall power for the FVR-55 to handle turbulence created by relative wind blowing over the superstructure of a
ship. A "pusher engine" is used to minimize contamination of sample air in flight by engine exhaust. Individual
payloads are integrated into a nose cone to allow for easy swapping of payloads between flights. Payloads are
powered at 12VDC from the plane with 200 W of power available.

Figure 1. FVR-55 with a) Clear Sky and b) Cloudy Sky payloads onboard.

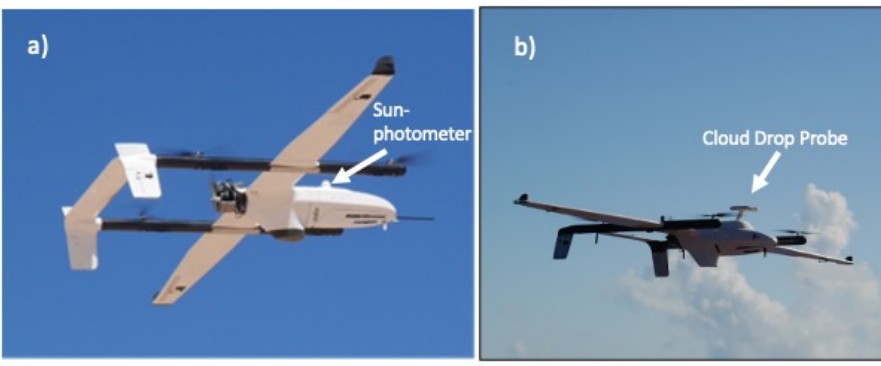


Table 1. Specifications of the FVR-55 UAS.

| Cruise speed | 25 m s$^{-1}$ |
|---|---|
| Weight with no payload or fuel | 20 kg |
| Maximum take-off weight | 29.5 kg |
| Endurance at maximum take-off weight including a 6.0 kg payload | 4.5 hrs |
| Altitude ceiling | 3,050 m |
| VTOL landing on land or ship | 6 m x 6 m recover area |
| Size | 4 m x 2.1 m x 0.3 m |





**2.2. Payloads**

An isokinetic inlet is mounted on the nose cone of the FVR-55 to bring sample air into the payload. The inlet has an
inline water trap with two outlets. One outlet is for the sample line, which is under vacuum. The larger outlet
exhausts any condensation to the atmosphere. Individual instruments sub-sample off of the sample inlet. A perma
pure drier is located downstream of the water trap and upstream of instruments in the nose cone. Instruments are
cooled in flight by air flow through vent shafts cut into the nose cone frame.

The data acquisition (DAQ) systems for the two payloads use different hardware and software but have the same
functionality. The Clear Sky payload DAQ is an Arduino based system that uses Labview software to collect data
and control the sensors. The Cloudy Sky payload uses a Raspberry PI running Python software to do the same. Both
DAQ systems collect and save data locally (on the aircraft) and also send data back to a ground station via a Silvus
radio link in near real-time. This communication link allows for command and control of the sensors during flight as
well as the ability to save a second copy of the data on the ground.

**2.2.1. Clear Sky Payload**

The Clear Sky Payload was designed to measure aerosol properties required for quantification of aerosol direct
radiative effects. All of the initial instruments in the payload were built by Brechtel Manufacturing Inc. (BMI,
Haywood, CA; ACCESS Model 9400) under a NOAA SBIR. The payload was first flown on a MANTA C1 UAS
from Ny-Ålesund, Svalbard, Norway in 2011 (Bates et al., 2013). The instruments included a mixing condensation
particle counter (MCPC) for measuring total particle number, or condensation nuclei (CN) concentration; a three-
wavelength Single Channel Tricolor Absorption Photometer (STAP) for measuring the aerosol light absorption
coefficient; and a multi-channel filter sampler for the collection of aerosol samples for post-flight chemical analysis.
Two instruments were added to the payload in 2014 including a Printed Optical Particle Spectrometer (POPS) for
the measurement of particle number size distribution (0.14 to 3 μm) (Telg et al., 2017) and a Mini Scanning Aerosol
Solar Photometer (mini-SASP) for the measurement of sun and sky radiance (Murphy et al., 2016). The payload also
includes Rotronic HC2-S3 and Innovative Sensor Technology (IST) HYT271 temperature and humidity sensors.
The updated version of the Clear Sky Payload was flown during a second campaign from Ny-Ålesund in 2015 (Telg
et al., 2017). A perma pure drier is plumbed into the sample line to provide dried air to the MCPC, STAP, and
POPS. The RH of the sampled air downstream of the drier was $34 \pm 1.6\%$, or ~ 8% lower than ambient RH, for
results reported here from a high-altitude flight off the coast of Oregon in August 2022. The Clear Sky Payload plus
the FVR-55 nose cone weighs 6 kg. The mini-SASP mounted on top of the FVR-55 nose cone is shown in Figure
1a. Table 2 lists the instruments in the Clear Sky Payload that were integrated into the FVR-55 nose cone. Further
details on each instrument are provided below. Comparisons between Clear Sky and bench top instruments are
presented in Sect. 3.





Table 2. Measured parameters and instrumentation in the Clear Sky Payload.

| Clear Sky Payload Instrumentation | | | |
|---|---|---|---|
| **Measured Parameter** | **Derivable Parameter(s)** | **Instrument** | **Uncertainty** |
| Total particle number concentration, > 0.005 μm | | Brechtel Mixing Condensation Particle Counter (MCPC) | ± 8%[a] |
| Particle number size distribution, 0.14 to 3 μm | Scattering coefficient, asymmetry parameter, Ångstrom exponent[b] | Portable Optical Particle Spectrometer (POPS) | ± 10% particle concentration accuracy |
| Aerosol light absorption coefficient (dry) (450, 525, and 624 nm) | Absorption aerosol optical depth ($AOD_{abs}$)<br><br>Single scattering albedo when paired with scattering coefficient derived from the measured number size distribution | Brechtel Single Channel Tricolor Absorption Photometer (STAP) | ± 33% at 1.0 $Mm^{-1}$,[c] |
| Sun and sky radiance (460.3, 550.4, 671.2, and 860.7 nm) | Aerosol optical depth (AOD) | Mini Scanning Aerosol Solar Photometer (mini-SASP) | 0.01 detection limit (AOD) |
| Chemical composition ($Na^+$, $NH_4^+$, $K^+$, $Mg^{2+}$, $Ca^{2+}$, $Cl^-$, $NO^3$, $Br^-$, $SO_4^{-2}$) | | Brechtel Multi-Channel Chemical Sampler | ± 5%[d]<br>± 8.5%[e] |
| T | | Rotronic HC2-S3 IST HYT271 | ± 0.1°C (<15 s)[f]<br>± 0.2°C (<15 s)[f] |
| RH | | Rotronic HC2-S3 IST HYT271 | ± 0.8% (<5 s)[f]<br>± 1.8% (< 4 s)[f] |

[a]Coincidence corrected concentration uncertainty at 10,000 $cm^{-3}$
[b]Using Mie theory
[c]Bates et al. (2013)
[d]Sample flow accuracy (uncertainty due to Chemical Sampler only)
[e]Overall uncertainty for the measurement of inorganic ions
[f]Time response

The MCPC (modified BMI Model 1710) has a 0.18 s response time, grows particles in a butanol-saturated flow, and
counts particles larger than 5 nm in diameter. Modifications to the butanol handling components of the commercial
Model 1710 were implemented to address the high vibration environment of the UAS (Bates et al., 2013). The
MCPC has a ± 8% coincidence corrected uncertainty for a particle concentration of 10,000 $cm^{-3}$.

The STAP provides real-time measurement of the aerosol light absorption coefficient at 450, 525, and 624 nm. Light
is transmitted from an LED source through a sample and a reference filter. The filter transmission is the ratio of the
signals from the two filters. The light absorption coefficient is proportional to the rate of decrease of light
transmittance divided by the flow rate of air through the filter (Bond et al., 1999). The raw data are averaged into 60
s values for the calculation of the rate of decrease of light transmittance. The minimum detectable level, MDL,
defined as the peak-to-peak noise with the instrument running particle free air, is 0.2 $Mm^{-1}$. Errors in the STAP
measurement include noise in the transmission value, uncertainty in the measured flow rate, and uncertainty in the
measured filter spot area (Anderson et al., 1999). A quadrature sum of these errors yields a relative uncertainty of ±





33% at 1.0 Mm$^{-1}$. In addition, light scattering by particles collected on the sample filter can lead to an
overestimation of absorption values by ~2% of the observed scattering coefficient (Bond et al., 1999). A correction
for scattering was not performed on the data collected in August 2022. The temperature and relative humidity of the
sample air flow drawn into the STAP was $12 \pm 1.6°C$ and $34 \pm 1.6\%$, respectively, for conditions encountered off of the
the coast of Oregon in August 2022.

The Brechtel Multi-Channel Chemical Sampler has eight filter holders (13 mm diameter) and a magnetically-driven
rotary valve manifold that distributes the vacuum from a central pump to each of the sampling channels. A remote
serial command is used to move the rotary valve to a new sampling channel in flight. The sample flow rate is 2.5 L
min$^{-1}$ which is measured by the pressure drop through a laminar flow element. One of the eight channels can be used
to maintain flow when a filter sample is not being collected. For the measurements reported here, filters were
extracted post-flight in a 17% methanol/water solution for analysis by ion chromatography (IC). Filters were
sonicated for 30 min. The filter extract was injected into a Metrohm 940 Professional IC Vario with 889 IC Sample
Center autosamplers and analyzed for inorganic cations ($Na^+$, $NH_4^+$, $K^+$, $Ca^{2+}$, $Mg^{2+}$) and anions ($Cl^-$, $NO_3^-$, $SO_4^{2-}$). A
Metrosep C6 - 100/4.0 column, 2 mMol HNO$_3$ eluent, and a flow rate 0.9 ml min$^{-1}$ were used for the cation analysis.
Metrosep A Supp 5 - 250/4.0 and Phenomenex Star Ion A300- 100/4.6mm columns in serial, a 1 mMol NaHCO$_3$
and 5 mMol Na$_2$CO$_3$ eluant, and a flow rate of 0.7 ml min$^{-1}$ were used for the anion analysis. Sources of uncertainty
in the chemical analysis include air volume sampled ($\pm 5\%$), the extract liquid volume ($\pm 3.5\%$), 2 times the
standard deviation of the blank, and precision/calibration of the method ($\pm 5\%$). Total average overall uncertainty
was $\pm 8.5\%$.

Particle number size distributions from 0.14 to 3.0 µm in diameter were measured with a POPS (Telg et al., 2017).
The POPS detects and sizes single particles based on the dependence of the scattering intensity on particle size. A
405 nm laser diode is used as a light source. The light scattering signal is collected at scattering angles between 38°
and 142° (Gao et al., 2016). As for the MCPC and the STAP, the temperature and RH of the sample air drawn into
the POPS was $12 \pm 1.6°C$ and $34 \pm 1.6\%$, respectively, for conditions encountered off of the coast of Oregon in
August 2022. Uncertainty for the POPs is ~10% of the total particle concentration.

Sun and sky radiance were measured with a miniSASP at wavelengths of 460.3, 550.4, 671.2, and 860.7 nm
(Murphy et al., 2016). Four independent telescopes, each with a unique interference filter, are housed in a single
aluminium block. A heater is integrated with a temperature controller to minimize condensation and keep the
photodiodes at an approximately constant temperature. The miniSASP scans the sky at the elevation angle of the
sun. A full azimuth revolution is made in about 30 s and measurements are made every 30 ms. Sun angle is
corrected for the tilt of the UAS. Each revolution of the miniSASP's telescopes results in a distinct peak
corresponding to the intensity of direct sun light. The aerosol optical depth of an atmospheric layer on the slant path
is the difference between the sun signal and Rayleigh scattering. Flight data from Svalbard in 2015 show a detection
limit better than 0.01 in AOD for a vertical profile through the bottom few kilometers of the atmosphere.



**2.2.2. Cloudy Sky Payload**

The Cloudy Sky payload was designed to characterize the relationship between cloud drop number concentration and particle number concentration and size below, within, and above cloud. The Cloudy Sky payload has a Brechtel miniature Scanning Electrical Mobility Sizer (mSEMS) for the measurement of particle number size distribution (0.01 to 0.3 μm) and total particle number concentration. A perma pure drier is plumbed into the sample line to provide dried air to the mSEMS. A miniature and light weight Cloud Drop Probe (DMT, CDP-2) is used to measure cloud droplet number concentration and size distribution between 2 and 50 μm. The payload also has Rotronic HC2-S3 and IST HYT271 temperature and humidity sensors. Cloudy Sky instrumentation and specifications are listed in Table 3. The Cloudy Sky payload was integrated into an FVR-55 nose cone in March 2021 at the L3Harris facility in Tucson, Arizona. The payload then flew three flights onboard the FVR-55 at the Florence Military Range up to an altitude of 2.6 km. The cloud droplet probe mounted on top of the FVR-55 nose cone is shown in Figure 1b. Further details on each instrument are provided below. Comparisons between Cloudy Sky and bench top instruments are presented in Sect. 3.

The Brechtel mSEMS (Model 9404) provided particle number size distributions for diameters between 0.01 to 0.3 μm every 30 seconds. Total particle number concentration was obtained by integrating the number concentration over the measured size distribution. The RH of sample air drawn into the mSEMS was $45 \pm 0.74\%$, which was ~ 40% below ambient RH, for the conditions encountered during flights off the coast of Oregon in August 2022. As for the Clear Sky payload, the MCPC has an $\pm 8\%$ coincidence corrected uncertainty for a particle concentration of 10,000 $cm^{-3}$.

A DMT CDP-2 was mounted on the top of a FVR-55 nose cone for measurement of cloud drop number concentration for diameters from 2 to 50 μm and retrieval of liquid water content.

Table 3. Measured parameters and instrumentation in the Cloudy Sky Payload.

| Cloudy Sky Payload Instrumentation | | | |
|---|---|---|---|
| **Measured Parameter** | **Derivable Parameter(s)** | **Instrument** | **Uncertainty** |
| Particle number size distribution and total number concentration (0.01 to 0.3 μm in diameter) | Scattering coefficient, asymmetry parameter, Ångstrom exponent[b] | Brechtel miniature Scanning Electrical Mobility Sizer (mSEMS) coupled with a MCPC detector | $\pm 8\%$[a] |
| Cloud droplet number concentration and size (2 to 50 μm) | Cloud liquid water content Cloud droplet effective diameter | DMT miniature Cloud Drop Probe (CDP-2) | |





| | | | |
|---|---|---|---|
| T | | Rotronic HC2-S3 IST HYT271 | ± 0.1°C[d] (<15 s)[b] ± 0.2°C (<15 s)[b] |
| RH | | Rotronic HC2-S3 IST HYT271 | ± 0.8%[d] (<5 s)[b] ± 1.8% (< 4 s)[b] |

219 [a]Coincidence corrected concentration uncertainty at 10,000 cm[-3]
220 [b]Response time
221

**3. Comparison of UAS and bench top measurements**

The degree of agreement between the bench and payload measurements of particle number concentration and absorption coefficient were evaluated by calculating the relative difference between the measurements as

$$relative\ difference\ = \left(\frac{x_{bench} - x_{uas}}{x_{bench}}\right) \qquad (1)$$

where $x_{bench}$ and $x_{uas}$ are the bench and UAS measured values, respectively. The overall experimental uncertainty was calculated as

$$experimental\ uncertainty = [(\delta x_{bench})^2 + [(\delta x_{uas})^2]^{1/2} / x_{bench} \qquad (2)$$

where $\delta x_{bench}$ and $\delta x_{uas}$ are the uncertainties in the bench and UAS measurements, respectively, as reported in Tables 1 and 2.

**3.1. Particle number concentration**

Particle number concentrations measured by the Clear and Cloudy Sky payloads and bench top instruments were compared during ATOMIC (The Atlantic Tradewind Ocean-Atmosphere Mesoscale Interaction Campaign), a cruise in the tropical North Atlantic (Quinn et al., 2021). The comparison took place on January 24, 2020 from 18:40 to 22:00 UTC. Sample air was drawn through a 5 m mast 18 m.a.s.l. and forward of the ship's stack. The mast was automatically rotated into the wind to maintain nominally isokinetic flow. Air entered the inlet through a 5-cm diameter hole, passed through a 7° expansion cone, and then into the 20-cm inner diameter sampling mast. The flow through the mast was 1 m³ min⁻¹. The transmission efficiency of the inlet for particles with aerodynamic diameters < 6.5 μm is greater than 95% (Bates et al., 2002). The bottom 1.5 m of the mast was heated so that the sample air was at an RH of 60 ± 5%. Stainless steel tubes extending into the heated portion of the mast were connected to bench top instrumentation and payload inlets with conductive silicone tubing.

A bench top MAGIC210 particle counter, which measures particles with diameters greater than 0.005 μm, was compared to the Clear Sky MCPC and the Cloudy Sky mSEMS (Figure 2, top of plot). Differential Mobility Particle Sizers (DMPS) and an Aerodynamic Particle Sizer (APS) were used for the comparison to the Clear Sky POPS for



particles with diameters greater than 0.14 μm (Figure 2, bottom of plot). A combination of an Aitken DMPS and an
Accumulation DMPS measures the size distribution between 0.002 and 0.8 μm in geometric diameter. The APS
measures the size distribution between 0.85 and 10.37 μm in aerodynamic diameter. The DMPS and APS size
distributions were merged by converting the APS data from aerodynamic to geometric values using calculated
densities and associated water mass at 60% RH based on the measured chemical composition (Quinn et al., 2002).
The DMPSs and APS are housed in a temperature-controlled box at the base of the inlet to maintain a uniform RH
across all instruments. Given that the payloads and bench instruments were measured from a common inlet and the
residence time in the tubing to the payloads was short, it is likely that RH differences in the sample air delivered to
the payload and bench instruments were negligible over the comparison period. The payload data were averaged into
5-minute time periods to match the DMPS/APS scan times.

The average difference between the bench MAGIC CPC and the Clear Sky payload MCPC number concentration
was $22 \pm 42$ cm$^{-3}$, resulting in an average relative difference of $5.2 \pm 0.86\%$. The relative difference is smaller than
the overall experimental uncertainty of $9.5 \pm 0.09\%$ indicating good agreement. The coefficient of determination, $r^2$,
for the comparison was 0.99. These results indicate that the trends in the two measures of number concentration
agreed well. However, the bench instrument was consistently higher by about 5%. Differences could be due to
particle losses in sampling lines.

The average difference between the bench MAGIC CPC and the Cloudy Sky integrated number concentration from
the mSEMS was $-1.9 \pm 9.8$ cm$^{-3}$, resulting in an average relative difference of $-0.19 \pm 0.67\%$. This difference is
smaller than the overall experimental uncertainty of $10.2 \pm 0.72\%$ indicating good agreement. A correlation between
the two measurements resulted in an $r^2$ value of 0.86. The mSEMS was, in general lower than the MAGIC CPC for
the first half of the comparison and higher for the second half, most likely due to changes in the mSEMS inversion
routine during the experiment.

The average difference between the bench DMPS/APS and the Clear Sky POPS for diameters greater than 0.14 μm
was $-11 \pm 7.6$ cm$^{-3}$, resulting in an average relative difference of $-31 \pm 6.7\%$. The overall experimental uncertainty
was $13 \pm 0.67\%$ indicating a systemic difference resulting in consistently lower values measured by the DMPS/APS
than the POPS, again likely associated with losses in sampling lines. The $r^2$ value for the correlation was 0.90.




*Figure 2. Comparison of particle number concentrations between bench and payload instruments for diameters*
*greater than 0.005 μm (top half of plot) and 0.14 μm (bottom half of plot) during ATOMIC on January 24, 2020.*
*Coefficients of determination, r², are for the regression between the payload CN concentration and the bench*
*instrument used for each size range.*

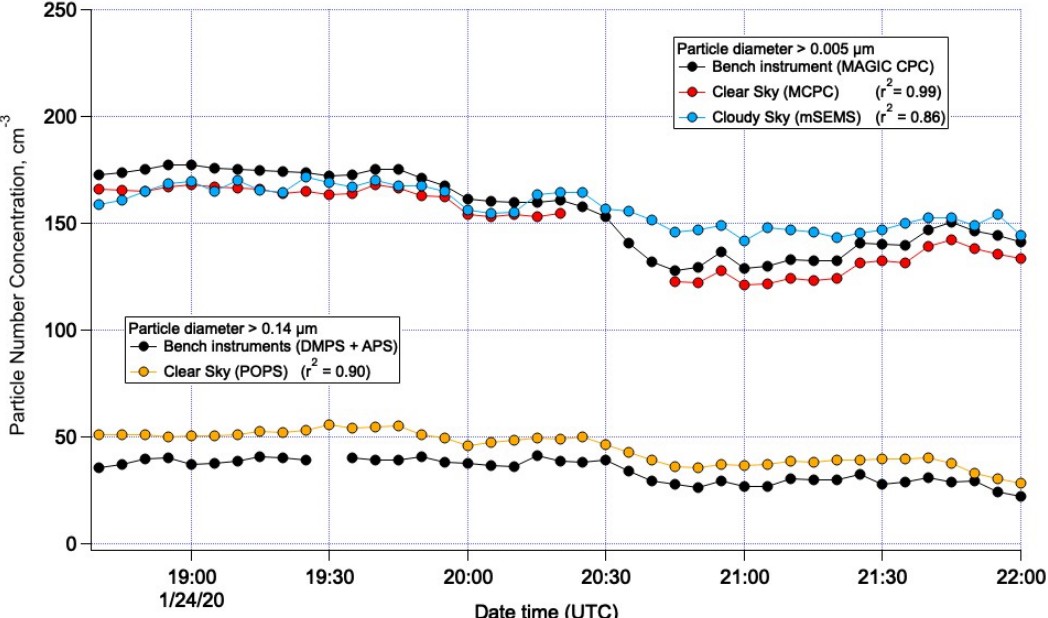



**3.2. Absorption coefficient**

The Clear Sky STAP (525 nm) was compared to a Radiance Research PSAP (530 nm) at PMEL on August 27, 2018
from 21:06 to 22:00 UTC. The 5 m mast described above was used to deliver sample air to the bench top PSAP and
to the Clear Sky payload. The bench PSAP was downstream of a Berner multi-jet cascade impactor with a 50%
aerodynamic cut-off diameter of 1.0 μm and a PermaPure nafion dryer (model PR-94). The Clear Sky STAP also
was downstream of a PermaPure nafion dryer so that both absorption signals were measured at < 25% RH. Data
were averaged to 30 s to minimize noise. A time series of the comparison and a correlation plot are shown in Figure
3a and b, respectively. The average absolute difference between the bench PSAP and the UAS STAP was 0.11 ±
0.34 Mm$^{-1}$. The average relative difference was 3.1 ± 12%, which was smaller than the overall experimental
uncertainty of 32 ± 3.9%. The r² value for the correlation was 0.81.




*Figure 3. Comparison of the Bench top PSAP and the Clear Sky UAS STAP as a) a time series and b) a*
*correlation plot. The comparison was done of ambient air at PMEL on 8/27/18.*

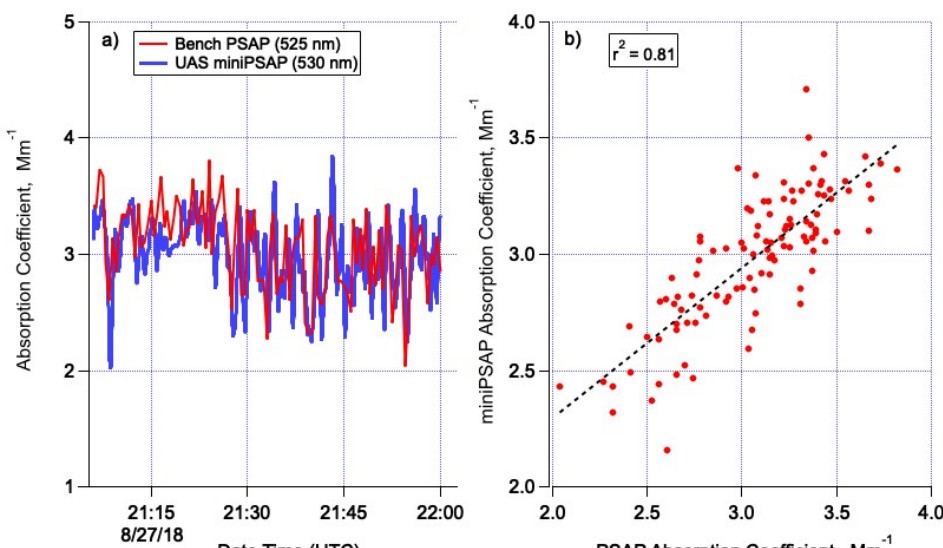



**3.3. Aerosol optical depth**

Aerosol optical depth (AOD) from the miniSASP was compared to a Solar Light Microtops during Flight 5 over the
Tillamook airport. The lowest altitude flown while the payload was still powered on before landing was 660 m. The
miniSASP measured at 550.4 nm and the Microtops at 500 nm. The miniSASP AODs were adjusted to 500 nm
using the Microtops-measured Ångstrom Exponent. Between 22:45 and 22:50 on August 11, 2022 the Microtops
AOD averaged $0.08 \pm 0.01$ while the miniSASP measured $0.07 \pm 0.02$ indicating agreement within overall
uncertainty. The lower average value for the miniSASP could be due to the higher altitude of the measurement. Due
to the limited period of comparison, further tests are warranted.

**4. Results**
**4.1. First shipboard flights**
The first shipboard flights of the FVR-55 with payloads onboard took place from March 9 to 11, 2022 from the
*TowBoatU.S. Richard L. Becker* off the coast of Key West, FL. A 6 m x 6 m launch pad was installed on the rear
deck to minimize interference with boat superstructure during take-off and landing (Figure 4). A total of 11 flights
were flown including 2 Functional Check Flights of the UAS, 4 with the Clear Sky payload, and 5 with the Cloudy
Sky payload. The first two flights were conducted 25 miles northwest of Key West with the remainder conducted 5
miles southeast of Key West. All flights were line-of-sight with a maximum altitude of 360 m due to the Certificate





of Authorization (COA) in place. Unfortunately, this low flight ceiling prevented clouds from being sampled. Table
4 provides a list of flights with duration, payload configuration, flight pattern, wind speed and ship heave. Wave
heights during all flights were observed to be between 0.3 and 0.6 m. Ship speed was 1 to 4 m s$^{-1}$.

Three flights occurred on March 10[th] and March 11[th] with both payloads being flown. With each payload in its own
nose cone, swapping of payloads between flights took 30 to 45 minutes. This time included readying the plane for
the next flight (installing fresh batteries and refueling).

*Figure 4. FVR-55 with the Clear Sky payload onboard on the 6 x 6 m launch pad on the rear deck of the*
*TowBoatU.S. Richard L. Becker.*

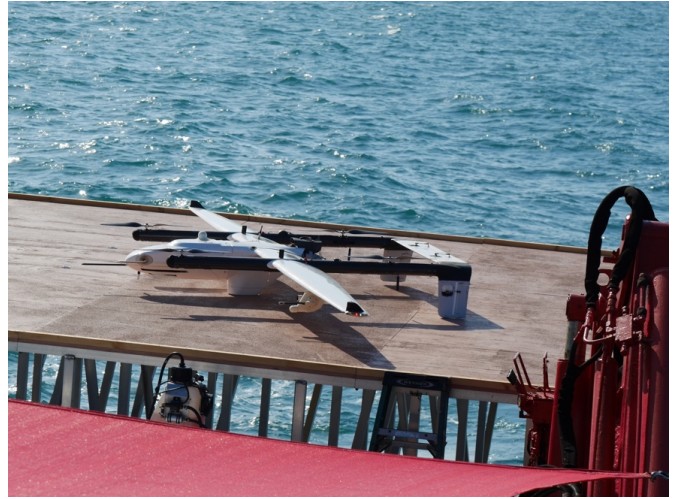



*Table 4. Shipboard flight information including duration, payload configuration, flight pattern, wind speed, and*
*ship heave.*

| Flight Number | Date | Duration (min) | Payload | Flight Pattern | Wind Speed (m s$^{-1}$) | Ship heave (m) |
|---|---|---|---|---|---|---|
| 1 | 3/9/23 | 21 | FCF[a] | | 7 | < 0.3 |
| 2[b] | 3/9/23 | 60 | Clear | Spirals between 60 and 335 m | | |
| 3 | 3/9/23 | 62 | Clear | Spirals between 60 and 335 m | 5.1 | < 0.3 |
| 4[c] | 3/10/23 | 5 | Cloudy | | 4.1 | 0.3 |
| 5 | 3/10/23 | 17 | FCF[a] | | 4.6 | 0.5 |
| 6 | 3/10/23 | 62 | Cloudy | Circles at 335 m | 4.9 | 0.5 |
| 7 | 3/10/23 | 60 | Cloudy | Circles at 335 m | 4.6 | 0.8 |
| 8 | 3/10/23 | 152 | Clear | Circles at 120 and 335 m | 2.6 | 0.3 |
| 9 | 3/11/23 | 183 | Cloudy | Circles at 335 m | 6.4 | 0.9 |
| 10 | 3/11/23 | 122 | Cloudy | Circles at 90, 150, 210, 270, and 335 m | 4.9 | 0.8 |



| 11 | 3/11/23 | 122 | Clear | Racetracks at 150 m | 5.6 | 0.5 |

[a]Functional Check Flight
[b]Telemetry file corrupted
[c]Generator failure, flight aborted

Examples of data collected during Clear and Cloudy Sky payload flights are shown in Figures 5 and 6, respectively.
The Clear Sky payload was flown on Flight 8. Initially four vertical profiles between 50 and 335 m were conducted
to identify the altitude of aerosol layers. Circles were then flown alternating between 335 and 120 m (Figure 5a).
Particle number concentrations decreased with height, ranging up to 5,000 cm$^{-3}$ at 120 m and decreasing to 1000 cm$^{-3}$
at 335 m for diameters greater than 0.005 μm (Figure 5b). Concentrations for diameters greater than 0.14 μm were
more than a factor of two lower. This result is expected given the large number concentration at diameters less than
0.2 μm. The flight track colored by time and the ship track are shown in Figure 5c. The ship travelled 1.3 km during
the flight. The plane landed within ± 0.36 m of the programmed spot on the launch pad.

*Figure 5. Data from the Clear Sky payload during Flight 8 offshore of Key West including a) altitude, b) total*
*particle number concentration for two size ranges (D$_p$ > 0.005 and 0.14 μm) and particle number size*
*distribution, and c) the flight track colored by time along with the ship track.*

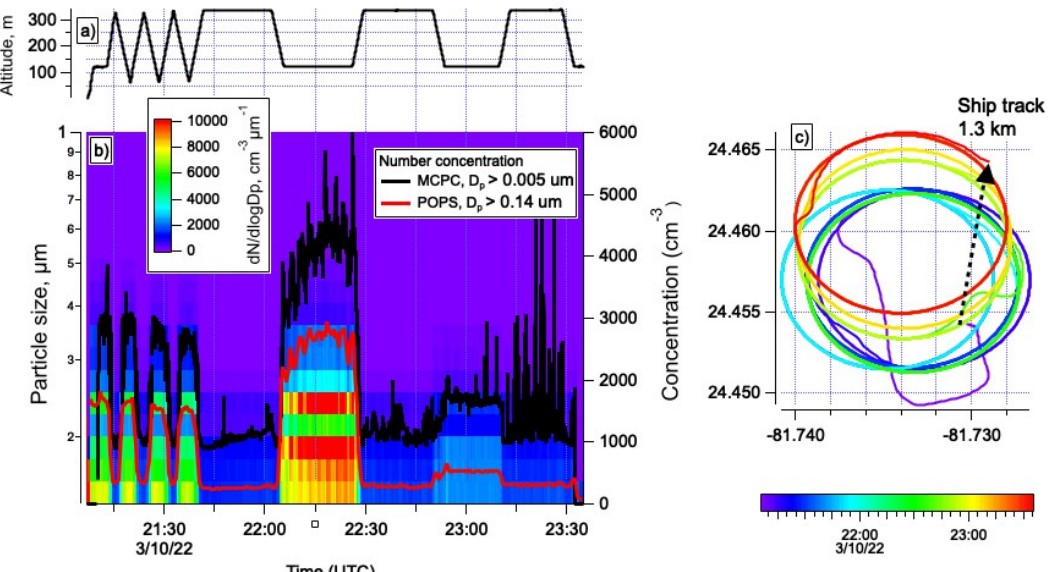



The Cloudy Sky payload was flown on Flight 6. After the initial ascent, circles were conducted at 335 m, the highest
altitude allowed by the COA in place (Figure 6a). Unfortunately, this altitude was below cloud bottom but the flight
served as a test of the aerosol instrumentation onboard. Initially particle number concentrations were around 3,000



cm$^{-3}$ but increased up to 8,000 cm$^{-3}$ after about 30 minutes of flight time (Figure 6b). As the number concentration
increased, the mean size of the particles shifted from about 0.12 to 0.16 μm. The flight track colored by time and the
ship track during the flight are shown in Figure 6c. The ship travelled 1.3 km during the flight with the plane landing
within ± 0.76 m of the designated spot on the launch pad.

*Figure 6. Data from the Cloudy Sky payload during Flight 6 offshore of Key West including a) altitude, b) total*
*particle number concentration and particle number size distribution, and c) the flight track colored by time along*
*with the ship track.*


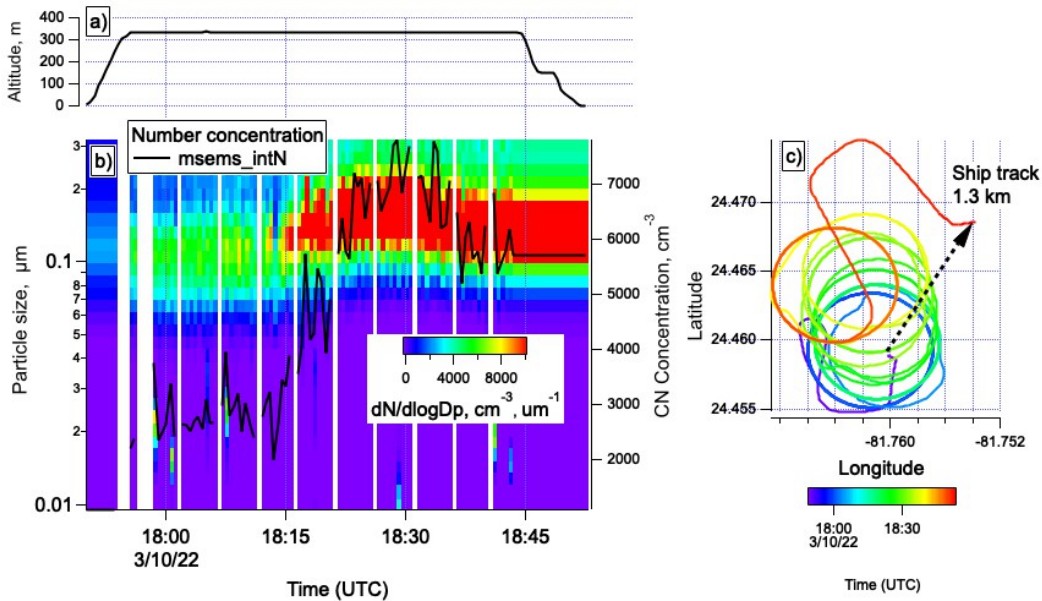



**4.2. First flights in cloud**

The FVR-55 with payloads onboard was flown from the Tillamook UAS Test Range (TUTR) in cooperation with
the Near Space Corporation (NSC) between August 9$^{th}$ and 17$^{th}$, 2022. TUTR is located at the Tillamook, OR airport
about 10 km from the coast. Flights were conducted over the airport and in offshore warning areas up to 40 km from
the airport under the NSC COA. Line of sight flights over the airport were conducted up to 1,370 m with the help of
visual observers. For the offshore BVLOS flights, a chase plane escorted the FVR-55 through non-controlled
airspace to the warning areas. NSC personnel communicated flights to the local FAA Air Traffic Control (Seattle
Center) and managed airspace de-confliction. Mission Control was based out of the Tillamook airport control tower.
The science team directed the Pilot-in-Control (PIC) to adjust flight tracks based on incoming, real-time data from



the payloads. Five flights with the Clear Sky payload and 9 flights with the Cloudy Sky payload were conducted for
a total of 38.5 flight hours (see Table 5).

*Table 5. TUTR flight information including duration, payload configuration, and flight pattern.*

| Flight Number | Date | Duration (min) | Payload | Flight Pattern | Comments |
|---|---|---|---|---|---|
| 1 | 8/9/12 | 120 | Cloudy Sky | Tracks below (300 m) and within (470 m) cloud[a] | Over airport |
| 2 | 8/9/12 | 123 | Cloudy Sky | Tracks below (400 m) and within (530 m) cloud[a] | Over airport |
| 3 | 8/10/12 | 118 | Cloudy Sky | Tracks below (610 m) and within (760 to 910 m) cloud[a] | Over airport |
| 4 | 8/10/12 | 119 | Cloudy Sky | Tracks below (610 m) and within (910 to 980 m) cloud[a] | Over airport |
| 5 | 8/11/22 | 213 | Clear Sky | Chase plane escort to offshore warning area for BVLOS[b] flights. Orbit in aerosol layer at 2550 m. | Offshore up to ~24 NM from airport |
| 6 | 8/12/22 | 169 | Cloudy Sky | Chase plane escort to offshore warning area for BVLOS[b] flights. Tracks below (800 m), within (1500 m), and above (2000 m) cloud. | Offshore up to ~24 NM from airport |
| 7 | 8/12/22 | 168 | Cloudy Sky | Tracks below (910 m) and within (1000 m) cloud[a] | Over airport |
| 8 | 8/13/22 | 78 | Cloudy Sky | Tracks below (1300 m) and within (1370 m) cloud[a] | Over airport |
| 9 | 8/14/22 | 151 | Clear Sky | Orbit in aerosol layer at 2300 m. | Over airport |
| 10 | 8/14/22 | 113 | Cloudy Sky | Chase plane escort to offshore warning area for BVLOS[b] flights. Clouds too far away to sample. | Over airport[c] |
| 11 | 8/15/22 | 223 | Cloudy Sky | Chase plane escort to offshore warning area for BVLOS[b] flights. PIC[c] handoff at Netarts Beach. Tracks below (300 m), within (400 m), and above (490 m) cloud. | Offshore up to ~16 NM from airport |
| 12 | 8/16/22 | 152 | Clear Sky | Chase plane escort to offshore warning area for BVLOS[b] flights. PIC[c] handoff at Bayocean Beach. Orbit in aerosol layer at 1800 m. | Offshore up to ~16 NM from airport |
| 13 | 8/16/22 | | | | Aborted. Chase plane issue. |
| 14 | 8/17/22 | 264 | Clear Sky | Orbit in aerosol layer at 1500 m. | Over airport |

[a]Above cloud flights were prevented by the line of sight COA over the airport.
[b]Beyond visual line of sight
[c]PIC = Pilot-In-Control

The track from Flight 5 with the Clear Sky payload onboard is shown in Figure 7 along with vertical profiles of RH,
particle number concentration, and aerosol absorption coefficient. The presence of an aerosol layer at ~2550 m is
clear based on increased particle number concentration and aerosol absorption. The factor of 4 increase in absorption
relative to values above and below the layer indicate the aerosol was likely made up of smoke. Results from the
filter sample collected in the aerosol layer show that non-sea salt potassium, a tracer of biomass burning, was
elevated at 0.04 μg m[-3]. HYSPLIT trajectory analysis indicates the sampled air mass was transported northward



along the Oregon coast where several fires were burning according to the NASA FIRMS (Fire Information for
Resource Management System) web site (https://firms.modaps.eosdis.nasa.gov/map/#t:adv;d:today;@-
117.1,41.0,6.0z).

*Figure 7. Flight 5 track from the TUTR colored by altitude (a) and vertical profiles of RH, particle number*
*concentration, and absorption coefficient (b).*

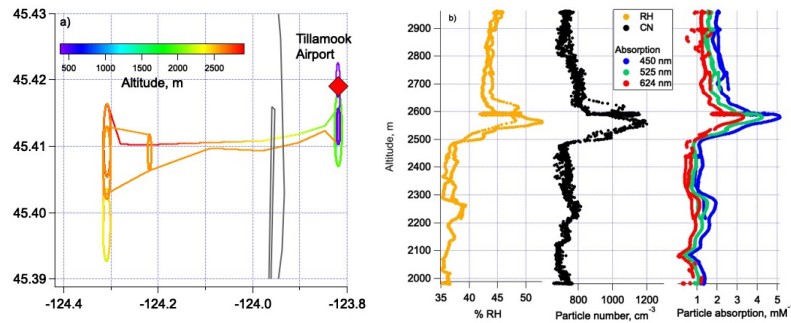


Vertical profiles of cloud drop number concentration for Flights 4, 6, and 11 are shown in Figure 8. All data points
are shown in grey and level leg averaged points are shown in red. Particle number concentrations for diameters
between 0.03 and 0.3 μm were derived from the integral of the mSEMS size distribution. A lognormal fit was
applied to the size distributions to extend the size range up to 2.0 μm to encompass the entirety of the accumulation
mode. The relationship between particle number concentration and cloud drop effective radius for the averaged level
leg data from the three flights is shown in Figure 9. Particle number concentration and cloud drop size were well
correlated ($r^2$ = 0.75 to 0.97) for all ranges of cloud liquid water contents sampled. An increase in particle number
concentration corresponded to a decrease in cloud drop size as expected for the first indirect or Twomey effect
(Twomey, 1977). Future data analysis will be done to explore relationships between aerosol number concentration
and size, cloud drop number concentration and size, and liquid water content for clouds at different altitudes.




*Figure 8. Vertical profiles of cloud drop number concentration for Flights a) 4, b) 6, and c) 11 from the TUTR.*
*All data points are shown in grey and level leg averaged points are shown in red.*

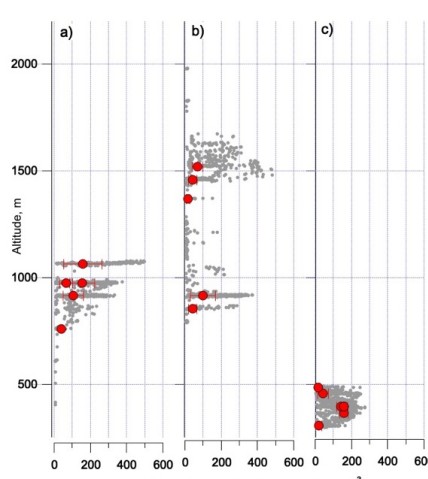

*Figure 9. Comparison of particle number concentration for diameters between 0.03 and 2.0 μm and cloud drop*

*effective radius averaged over the altitude level leg data shown in Figure 8. Data are binned by ranges of cloud*

*liquid water content.*

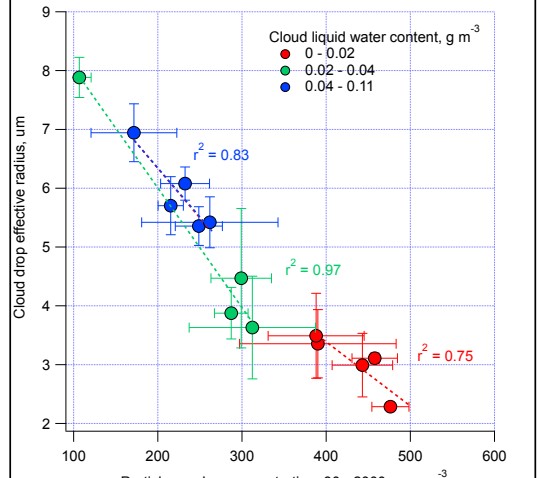



## 5.0. Conclusions

The initial results described here indicate that the FVR-55 UAS with Clear and Cloudy Sky payloads onboard offers a unique platform for observations relevant to aerosol direct and indirect radiative forcing. This observing platform is deployable at sea with less cost and greater flight frequency than a crewed air craft. Potential applications of this technology extend beyond aerosol – cloud observations to marine mammal assessments, harmful algal blooms, and radiative impacts from forest fires.

Next steps include upgrading to the larger L3Harris Fixed Wing VTOL Rotator, the FVR-90. The Clear and Cloudy Sky payloads will be integrated into FVR-90 nose cones which will allow for the addition of instruments and longer flight endurance. The planned added instruments include upward and downward looking pyranometers to assess direct connections between particle number and concentration, cloud drop concentration and size, and radiation. In addition, instrumentation will be added to both payloads for the measurement of particle number size distributions from 5 nm to 3 $\mu$m. Although larger, the FVR-90 is operable from a ship thereby fulfilling the need of aerosol, cloud, and radiation measurements in the marine atmosphere.

**Data availability**

Flight data are publicly available at https://saga.pmel.noaa.gov/Field/Tillamook2022/.

**Author contributions**

PKQ and TSB designed the experiments. DJC and JEJ built and operated the payloads. LMU analyzed the chemical data. PKQ prepared the manuscript with contributions from all co-authors.

The authors declare that they have no conflict of interest.

**Acknowledgements**
We thank Aaron Farber and the entire L3Harris Latitude Engineering team for developing, fabricating, and flying the FVR-55. We thank the caption and crew of the TowBoatUS Richard L. Becker and Chuck Bagnato and Eric Waters from the Tillamook UAS Test Range for their contributions toward successful flights. We thank Alexander Smirnov of NASA GSFC for the microtops calibration, processing, and data quality assurance. We also thank NOAA OMAO and UxSRTO for logistical and financial support. This is PMEL contribution number 5537.

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
