# Peer review of "Use of an Uncrewed Aerial System to Investigate Aerosol Direct and Indirect Radiative"

_EGUsphere, 2023_

## Author Comment (AC1)

Reviewers' comments are in black. Responses are in blue with changes to the text of the paper in blue italic. All line numbers in the responses refer to the revised version, not the original.

**Responses to Reviewer 1**

An uncrewed aerial system (UAS) that developed for observations of aerosol and cloud properties in the marine atmosphere was introduced in this study. Compared to the regular UAS designed for observation of vertical aerosol and cloud properties, the new UAS (Fixed Wing VTOL Rotator or FVR-55), reported to have the advantage of much longer endurance (~4 hours), much higher height ceiling (~3 km), with the ability of carrying heavier payloads (~6 kg). As the Payload equipped with commercialized instruments, the technological advances could be refer to the FVR-55 and its sampling system connected to the Payload, however, details of which was not provided clearly. In addition, the observation data was not well analyzed and weakened its credibility. Thus, before its publication, the following issues should be properly revised and improved.

Specific and technical comments:

Line 85, by using the piston engine and liquid fuel to supply power for fixed wing flight, the engine exhaust do affect the sample air in flight, especially for the cycles flight pattern. Had the authors evaluated such influence? In addition, why the "pusher engine" could be minimize the contamination of sample air from engine exhaust?

With any UAS using a gas engine there will be some exhaust that can potentially contaminate the sampled air. A pusher engine, however, reduces this contamination by exhausting behind the UAS while the UAS is moving forward. This approach does not always work, as the reviewer points out, for example when the flight track includes circles or spirals. Engine exhaust is readily identifiable by short-lived increases in particle number concentration. We removed all data during these contaminated periods. We have added the following text to the paper (lines 87 – 90):

*"A "pusher engine" is used to minimize contamination of sample air in flight by exhausting the engine aft while the UAS flies forward. When the flight track includes circles or spirals engine contamination can occur but is readily identifiable by short-lived bursts in particle number concentration. We removed all data during these contaminated periods."*

Line 97-101, It is unclear how the sample air passed through the nose cone of the FVR-55 then bring into the payload? Figure that showing the internal structure of the nose cone and the sample lines is needed here, which is important to evaluated the particle loss in the sample lines.

We have added more information about the plumbing inside the payloads as well as a flow diagram to Section 2.2.1. as follows (lines 104 – 134):

*"An isokinetic inlet is mounted on the nose cone of the FVR-55 to bring sample air into the payload under vacuum (See Figure 2). No changes in particle number concentration coinciding with the UAS transitioning from large spirals (1 to 2 km) to level leg flights were observed, indicating the performance of the isokinetic inlet was not impacted by a spiral flight pattern. Since particle number concentrations are dominated by the submicron size range this metric does not rule out effects in supermicron size ranges. In addition, the slow air speed of the UAS (25 m sec$^{-1}$) is expected to decrease impacts of the flight pattern on transmission of submicron particle through the inlet into the payload.*

*Sample air first encounters an inline water trap where water droplets are removed through impaction. The water trap has two outlets -- one outlet is for the sample line, which is under vacuum. The larger outlet exhausts condensate through a drain line that also allows excess ram air to passively exit the sampling system. Individual instruments sub-sample off of the sample inlet. For the Clear Sky payload, a perma pure drier is located downstream of the water trap and upstream of all instruments except the filter sampler (Figure 2a). For the Cloudy Sky payload, a perma pure drier is located downstream of the water trap and upstream of the mSEMS (Figure 2 b). A restricting orifice and filter on the inlet of the perma pure sheath air combined with a vacuum on the outlet of the sheath air was used to remove moisture from the sample stream. Instruments are cooled in flight by air flow through vent shafts cut into the nose cone frame."*

*Also, Figure 2 was added to show flow diagrams for both payloads. See below.*

**Figure 2. Flow diagrams for a) Clear Sky and b) Cloudy Sky payloads.**

[Figure]

[Figure]

Line 124-125, did the perma pure drier used here need the sheath air to take away the wet purge gas?

The following text has been added to Section 2.2.1. (lines 119 - 121):

*"A restricting orifice and filter on the inlet of the perma pure sheath air combined with a vacuum on the outlet of the sheath air was used to remove moisture from the sample stream."*

Line 159-162, is the multi-channel filter sampler share the same sampling line with the other instruments? If so it would compete the air mass with the MCPC and POPS which has much lower flow rate. Please explain it.

The new flow diagrams in Figure 2 (see above) show the split in the sample air inlet between the chemical sampler and the real-time instruments indicating they did not share the same flow path.

In addition, what kinds of filter was used and what about the background concentration of the mentioned elements? Considering the relative low sampling flow and limited sampling time (few hours) in flight, the collected particle mass in the filter would be insufficient for the analysis of chemical species like the water soluble ions.

The ability to detect soluble ions with this sampling method depends on the volume of liquid used for extraction of the filters and the aerosol loading in the sampled air. Obviously more ions are detectable in fire plumes (as was the case reported here) than in clean remote marine air. We have added the following text to the paper:

*Lines 200 - 201: "13 mm Millipore Fluoropore 1.0 μm PTFE membrane filters were used for sample collection."*

*Lines 202 - 203: "The volume of liquid used to extract the filters was minimized to 1 mL to increase the sensitivity of the method."*

*Lines 210 - 211: "Only ion concentrations above 2 times the standard deviation of the filter blank are reported here."*

Line 212-213, please provided the information on how the liquid water content was retrieved here.

The following text has been added (lines 258 - 264):

*"Liquid water content was derived from the cloud droplet size distribution provided by the DMT CDP-2. The CDP-2 measures cloud droplet counts and sizes them into 30 bins from 2-30 um. The count in each bin is converted to a concentration using the cross-sectional surface area of the sensing beam (0.24 um$^2$) and the speed of the aircraft to*

*determine the volume of air sampled per second. Once the concentration is known, the volume of cloud droplets per volume is calculated and converted to mass per volume assuming a density of 1.0."*

Line 246-247, the RH in the sample air on the bench top measurement (~60%) is different from that those in the Clear Sky payload and the Cloudy Sky payload for comparison, which could be an important factor for the difference in measurements. Please add discussion about the influence of RH in the sample air.

The following statement is in the submitted paper (Lines 314 - 316) "Given that the payloads and bench instruments were measured from a common inlet and the residence time in the tubing to the payloads was short, it is likely that RH differences in the sample air delivered to the payload and bench instruments were negligible over the comparison period". We have added the following text to the paper (lines 296 - 297) to further clarify that a common inlet was used:

*"For both the payload and the bench top instruments, sample air was drawn through a 5 m mast 18 m.a.s.l. and forward of the ship's stack."*

Line 268-269, Line 281, the author suggested the systemic difference in particle number concentration measurement likely due to the particle losses in sampling lines, could the author provide some quantitative analysis results about the particle losses?

Quantification of particle losses is beyond the scope of this paper but planned for the future. We have added the following text to the paper (Line 324):

*"Particle losses will be quantified in future experiments."*

Line 387, Table 5, During the TUTR fights, How does the author determine if the FVR-55 is inside a cloud, or under a cloud?

We have added text to the caption of Table 5 stating that (Lines 469 - 470):

*"Time within cloud is based on a measured cloud drop number concentration above 5 cm$^{-3}$."*

Line 411-415, the relationship between particle number concentration and cloud drop size showed in Figure 9 is interesting. More in-depth analysis is suggested here, at least, discussion about the different correlations under different cloudy liquid water content is needed.

As pointed out in the text (lines 498 - 99), further data analysis is required to explore relationships between aerosol number concentration and size, cloud drop number concentration and size, and liquid water content for clouds at different altitudes. These analysis are beyond the scope of this paper.

---

## Author Comment (AC2)

**Responses to Reviewer 2**

Reviewers' comments are in black. Responses are in blue with changes to the text of the paper in blue italic. All line numbers in the responses refer to the revised version, not the original.

General comments:

This paper provides an overview of the engineering effort to operate a UAS for atmospheric observation from a ship and the Tillamook UAS test range. The capability development is very exciting and shows great potential for using a UAS to support the Marine atmosphere study. The paper is well written. There are excessive efforts involved with the UAS program development. However, the scientific aspect of this paper can be strengthened with major revision. The main concerns are listed below:

1.   The manuscript didn't provide enough detail about the isokinetic inlet system. This inlet is the most critical component to ensure representative aerosol collection.

We have added a more complete description of the isokinetic inlet in Section 2.2.1. (lines 105 – 112):

*"An isokinetic inlet is mounted on the nose cone of the FVR-55 to bring sample air into the payload under vacuum (See Figure 2). No changes in particle number concentration coinciding with the UAS transitioning from large spirals (1 to 2 km) to level leg flights were observed, indicating the performance of the isokinetic inlet was not impacted by a spiral flight pattern. Since particle number concentrations are dominated by the submicron size range this metric does not rule out effects in supermicron size ranges. In addition, the slow air speed of the UAS (25 m sec⁻¹) is expected to decrease impacts of the flight pattern on transmission of submicron particle through the inlet into the payload."*

2.   Experimental design issues
    1.   It is unclear how to sample the aerosol during a cloud flight. Is there a CVI inlet? How do you prevent the small droplet from getting into the inlet and ensure only the aerosol, not small droplets passes through?

The following text has been added to Section 2.2.1. (lines 114 - 116):

*"Sample air first encounters an inline water trap where water droplets are removed through impaction. The water trap has two outlets -- one outlet is for the sample line, which is under vacuum. The larger outlet exhausts condensate through a drain line that also allows excess ram air to passively exit the sampling system."*

    2.   How does the aerosol sampling behave during the spiral flight pattern? Does the isokinetic inlet work properly? Usually, the isokinetic inlet works well during a leveled flight leg only.

See the response to comment 1 above.

    3.   When the aircraft is circling at one altitude, how do you prevent sampling the aircraft exhaust?

Engine exhaust is readily identifiable by short-lived increases in particle number concentration. We removed all data during these contaminated periods. We have added the following text to the paper (lines 87 – 90):

*"A "pusher engine" is used to minimize contamination of sample air in flight by exhausting the engine aft while the UAS flies forward. When the flight track includes circles or spirals engine contamination can occur but is readily identifiable by short-lived bursts in particle number concentration. We removed all data during these contaminated periods."*

Specific comments:

Abstract: this UAS capability development is essential to ensure the success of the scientific study. The abstract doesn't emphasize its importance. Although the data and results are limited, there are many lessons learned that should be shared.

The following text has been added to the abstract (lines 19 – 21):

*"The development of this UAS technology for flights from ships and coastal locations is expected to greatly increase observations of aerosol radiative effects in the marine boundary layer over both temporal and spatial scales."*

Section 2.2, How was the isokinetic inlet controlled? Passive or active? Please provide the characteristics of the performance and operation ranges.

Pleases see the response to comment 1 above about control of the inlet. The performance and operation ranges of the isokinetic inlet have not yet been fully characterized. Future wind tunnel tests are planned for this purpose. We have added the following text (lines 111 - 112):

*"Wind tunnel tests are planned for the determination of the particle passing efficiency as a function of air speed and particle size."*

Section 2.2.1, what is the sample rate for this payload? 1 Hz?

The following text has been added to the description of the Clear Sky payload (now in Section 2.2.2.) (lines 160 – 163):

*"Sampling rates were 1 sec for all real time instruments while filter samples were collected over a period of minutes to hours."*

Line 159-172, what is the detection limit for the chemical analysis? How long will the flight last to provide reasonable chemical composition data?

We define the detection limit for the chemical analysis as 2 times the standard deviation of the blanks. Flight duration needed to be above detection limit depends on the aerosol loading being sampled. Obviously more ions are detectable in fire plumes (as was the case reported here) than in clean remote marine air.

We have added the following text to the paper:

*Lines 210 - 211: "Only ion concentrations above 2 times the standard deviation of the filter blank are reported here."*

Section 2.2.2, How does the mSEMS sample the ambient aerosol? RH range? What is the mSEMS operating condition? Such as flowrates, sampling rate, and scanning cycle?

We have added the following text to better describe the mSEMS measurements in the Cloudy Sky payload (Section 2.2.3., lines 236 – 239):

"The RH of the sheath air was measured during operation. The RH of the dried sample air depended on ambient conditions but ranged from 35 to 45% for the flights from Tillamook. Operating conditions for the mSEMS included a sheath flow rate of 2.5 lpm, sample flow rate of 0.36 lpm, and a size scan of 30 bins at 1 sec per bin resulting in a sampling rate of 30 sec for each size distribution."

Line 234, how do you determine the uncertainties in the bench and UAS measurements for this study? From literature?

The following text has been added to Section 3 (line 283):

*"...the uncertainties in the bench and UAS measurements, respectively, as reported in Tables 1 and 2 and taken from manufacturer specifications."*

Line 256 -258, What are the density and chemical composition values used with this study? From the in situ measurements or literature from 2002?

Text has been changed to (line 312):

*"...based on the range of measured chemical compositions reported by Quinn et al. (2002)."*

Line 264-266, please double-check the precision in the percentage. Can you really get +-0.86% variance?

The text has been changed to *"0.9%"* (line 320).

Line 271, what size range is used for the Cloudy Sky integrated number concentration? How does it compare with the Magic CPC?

As shown in Figure 3, the integrated number concentrations for diameters greater than 0.005 um from the MAGIC CPC and the Cloudy Sky integrated number from the mSEMs were compared. $R^2$ for the comparison was 0.86.

Line 278, again, what is the size range used for the DMPS/APS compared to the Cloudy Sky POPS?

Figure 3 also shows the comparison between the DMPS/APS and the Clear Sky POPS for diameters greater than 0.14 um. $R^2$ for the comparison was 0.90.

Fig 2, why not include a similar 1:1 plot as Fig. 3b?

The 1:1 plot for the bench PSAP and UAS STAP comparison was shown because of the inherent noise in low levels of absorption. The time series in Figure 3 (formerly Figure 2) have much less noise making the comparison clearer so that a 1:1 plot was not needed.

Fig 3, Does this plot compare PSAP and STAP or PSAP with miniSASP? Some errors with the labels and legend.

Thank you for pointing this out. The legend and axis labels have been corrected.